# The expression of stem cells markers and its effects on the propensity for recurrence and metastasis in bladder cancer: A systematic review

**Agus Rizal Ardy Hariandy Hamid**[ID]*, **Yasmina Zahra Syadza, Oliver Emmanuel Yausep, Roberto Bagaskara Indy Christanto, Bayu Hernawan Rahmat Muharia**[ID]**, Chaidir Arif Mochtar**

Department of Urology, Faculty of Medicine Universitas Indonesia–Dr. Cipto Mangunkusumo Hospital, Jakarta, Indonesia

* rizalhamid.urology@gmail.com

**Data Availability Statement:** All relevant data are within the paper and its Supporting Information files.

## Abstract

Bladder cancer is one of the most frequent cancers of the urinary tract, associated with high recurrence rates and metastasis. Cancer stem cells (CSCs) are a subpopulation of cancer cells characterized by high self-renewal and differentiation capacities, resulting in increased cancer recurrence, larger tumor size, higher rates of metastasis, higher resistance to treatment, and overall poorer prognosis. This study aimed to evaluate the role of CSCs as a prognostic tool to predict the risks of metastasis and recurrence in bladder cancer. A literature search was conducted across seven databases from January 2000 to February 2022 for clinical studies investigating the use of CSCs to determine the prognosis of bladder cancer. The following keywords were used: ("Bladder Cancer" OR "Transitional Cell Carcinoma" OR "Urothelial Carcinoma") AND ("Stem Cell" OR "Stem Gene") AND ("Metastasis" OR "Recurrence"). A total of 12 studies were deemed eligible for inclusion. SOX2, IGF1R, SOX4, ALDH1, CD44, Cripto-1, OCT4, ARRB1, ARRB2, p-TFCP2L1, CDK1, DCLK1, and NANOG, which were all identified as CSC markers. Several of these markers have been implicated in the recurrence and metastasis of tumor in bladder cancer, which played a role as prognostic factor of bladder cancer. Given the pluripotent and highly proliferative properties of CSCs. CSCs may play a role in the complex biological behavior of bladder cancer, including, but not limited to, its high rates of recurrence, metastasis, and resistance to treatment. The detection of cancer stem cell markers offers a promising approach in determining the prognosis of bladder cancer. Further studies in this area are thus warranted and may contribute significantly to the overall management of bladder cancer.

## Introduction

Bladder cancer (BCa) is the most common neoplasm of the urinary tract and the fifth most prevalent malignancy worldwide. High-grade bladder tumors are more likely to progress to

**Funding:** This work was supported by Universitas Indonesia for funding this research through PUTI Grant with contract number NKB-1533/UN2.RST/HKP.05.00/2020 to ARAHH. The grant did not play any role in the study design, data collection and analysis, decision to publish, or preparation of the manuscript.

**Competing interests:** The authors have declared that no competing interests exist.

muscle-invasive disease and have a higher tendency to undergo distant metastasis. In contrast, low-grade tumors rarely invade the bladder musculature and metastasize [1–3]. However, both non-muscle-invasive bladder cancer (NMIBC) and muscle-invasive bladder cancer (MIBC) have a high propensity of recurrence, with a 50–90% probability of recurrence within five years [4]. In addition, metastatic BCa is considered incurable [5]. Several established risk factors related to higher risks of disease progression include tumor grade, tumor size, and tumor multiplicity; however, these risk factors are insufficient to address important prognostic indicators such as recurrence rates and progression in individual cases [6, 7]. Thus, there is an urgent need to identify novel, more reliable prognostic factors for BCa.

Cancer stem cells (CSCs), also known as tumor-initiating cells, are a subpopulation of undifferentiated yet tumorigenic cells within a neoplasm that are capable of tumor initiation, self-renewal, and proliferation, which are thought to be responsible for tumor progression, relapse, metastasis, and heterogeneity [8, 9]. CSC expressions have been identified in multiple human solid tumors, including breast, prostate, ovarian, and lung cancers, and are significantly associated with metastasis-free survival and other clinical outcomes [10]. Bladder cancer stem cells (BCSCs) were first identified using markers for isolation of normal stem cells in 2009 [11]. Since then, BCSCs have emerged as a growing field of research, with genome-wide screening methods and platforms for establishing therapeutic targets for tumor-initiating cell populations [12]. A more profound understanding of BCSCs and their effects on BCa may provide helpful prognostic tools and novel therapeutic targets. However, the clinical impacts of BCSC expressions and functions have not been fully elucidated yet. Hence, this systematic review aims to evaluate all available evidence regarding BCSCs and their roles in predicting the risks of metastasis and recurrence in BCa.

## Materials and methods

### Objectives

This article aims to provide a systematic review of primary clinical studies to identify the BCSCs markers, which have played a role as prognostic factors in BCa patients.

### Study design

This systematic review was created in accordance with the guidelines for Preferred Reporting Items for Systematic Reviews and Meta-analysis (PRISMA) [13]. We determined inclusion criteria, data synthesis methods, and outcomes in advanced in a protocol registered with PROSPERO (CRD42021268964).

### Search strategy

A literature search for clinical studies evaluating CSCs as a prognostic indicator in BCa published from January 2000 to February 2022 was conducted from several databases, such as Pubmed, Scopus, EMBASE, Science Direct, Proquest, CINAHL, and The Cochrane Library. The search following key terms used based on the PICO were applied to identify eligible publications: ("Bladder Cancer" OR "Transitional Cell Carcinoma" OR "Urothelial Carcinoma") AND ("Stem Cell" OR "Stem Gene") AND ("Metastasis" OR "Recurrence"). Initially, study titles and abstracts were screened. Subsequently, full text analysis of selected articles was done based on pre-set eligibility criteria. In addition, the reference lists of included studies were further evaluated to identify potential studies. Literature screening and analysis were undertaken separately by two independent researchers.

### Eligibility criteria

Our inclusion criteria were as follows: (1) studies evaluating the impacts of BCSC expressions on BCa recurrence and/or metastasis; (2) prospective or retrospective cohort studies or case control studies; and (3) publications written in English. Animal and *in vitro* studies were excluded.

## Outcomes

The primary outcomes of this systematic review are the effect of BCSC expressions toward recurrence and metastasis in BCa. In addition, survival analyses from several studies were also presented.

### Data extraction

Two authors independently run the systematic search and screened the articles. From all eligible studies, data were also extracted independently, and any disagreement were resolved through discussion among all authors. Data recorded from each study were as follow: author's name, year of publication, study design, number of study's participant, intervention given to the participant, method used for gene expression analysis, outcomes (recurrence-free survival and metastasis-free), and mean or median year follow-up. The effect measures used were the hazard ratios and their respective 95% confidence intervals for both univariate and multivariate analyses.

### Quality assessment

The qualities of the selected studies were assessed using the Newcastle-Ottawa Scale (NOS). Using this tool, selected studies were assessed based on three aspects: the selection of the study groups; the comparability of the groups; and the ascertainment of either the outcome of interest for case-control or cohort studies, respectively. Good quality studies have 3 to 4 stars in the selection component, 1 to 2 stars in the comparability component, and 3 stars in the outcome component. Fair quality studies have 2 stars in the selection component, 1 to 2 stars in the comparability component, and 2 to 3 stars in the outcome component. Poor quality studies have 0 to 1 star in the selection component, 0 star in the comparability component, and 0 to 1 star in the outcome component. The quality assessment showed in Table 1.

## Result

### Study selection

The flow diagram in the form of PRISMA diagram for study selection is shown in (Fig 1). We included 12 clinical studies evaluating effects of BCSCs expression on tumor recurrence and/or metastasis, consisted of cohorts and case controls studies, involving at least 2230 patients (one study did not specify the sample size) with BCa and 68 non-tumor tissue for control in this systematic review. All the eligible studies were published between 2012 and 2020.

### Study characteristics

All of the studies selected were reviewed and the result were displayed on Table 1. We reviewed the intervention given to the patient, gene expression analysis, and outcomes which was consisted of recurrence and metastasis. Eleven out of 12 studies assessed the recurrence-free survival related to the BCSCs and five studies out of 12 studies assessed the metastasis associated

**Table 1. Risk of bias assessment using Newcastle Ottawa Score (NOS).**

| No | Study | Selection (Max *) | | | | Comparability (Max **) | Outcome (Max *) | | | Score |
|----|-------|------------------------------|------------------------------|--------------------------|-------------------------------|----------------------------------------------------|------------------------|-----------------------------------------------------|---------------------------------|-------|
| | | Representativeness of exposed cohort | Selection of exposed cohort | Ascertainment of exposure | No outcome of interest at start | Comparability of cohorts based on design or analysis | Assessment of outcome | Was follow up long enough for outcomes to occur | Adequacy of follow up of cohorts | |
| 1 | Ruan et al, 2012 [6] | * | * | * | * | * | * | * | * | 8 |
| 2 | Keymoosi et al, 2014 [13] | * | * | * | * | * | * | * | * | 8 |
| 3 | Wei et al, 2015 [14] | * | * | * | * | * | * | * | * | 8 |
| 4 | Senol et al, 2015 [15] | * | * | * | * | * | * | * | * | 8 |
| 5 | Sedaghat et al, 2016 [2] | * | * | * | * | * | * | N/A | * | 7 |
| 6 | Siddiqui et al, 2019 [16] | * | * | * | * | * | * | * | * | 8 |
| 7 | Chiu et al, 2020 [17] | * | N/A | * | * | N/A | * | N/A | * | 5 |
| 8 | Shen et al, 2015 [1] | * | * | * | * | ** | * | * | * | 9 |
| 9 | Xu et al, 2015 [18] | * | * | * | * | * | * | * | * | 8 |
| 10 | Heo et al, 2020 [19] | * | * | * | * | * | * | N/A | * | 7 |
| 11 | Shaifei et al, 2019 [20] | * | * | * | * | * | * | N/A | * | 7 |
| 12 | Kallifatidis et al,2019 [21] | * | * | * | * | * | * | * | - | 7 |

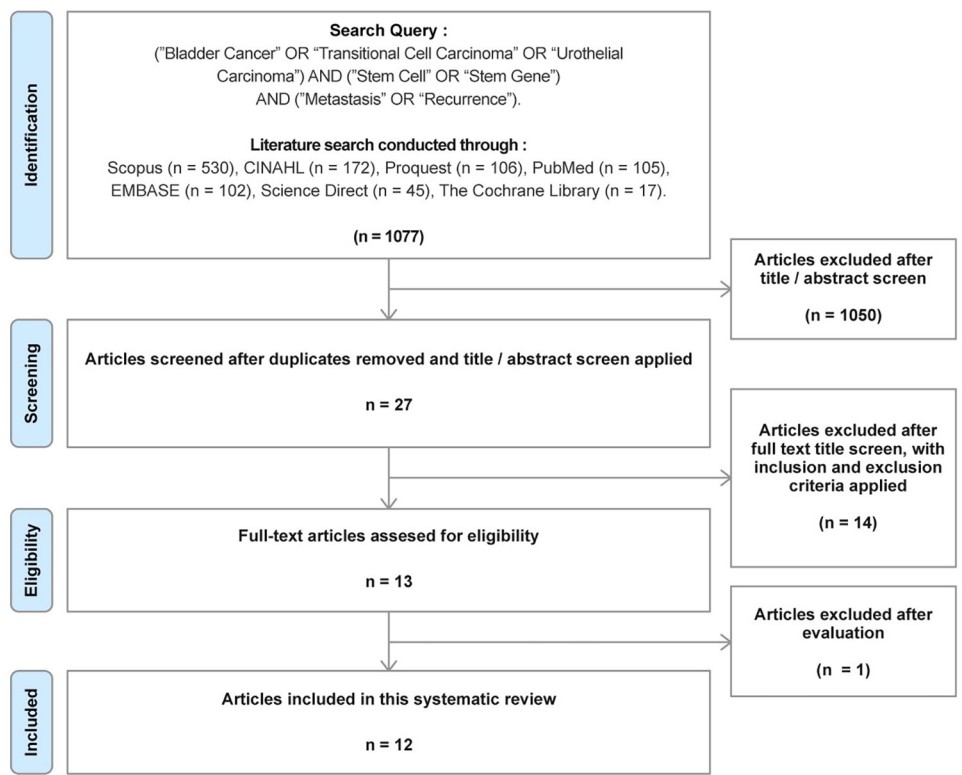

**Fig 1. Study Selection using PRISMA flow diagram.** Articles were identified and screened for eligibility, 12 clinical studies were included.

with BCSCs. Only four studies which analyzed both tumor recurrence and metastasis. Only three studies provided data about mean or median follow-up time Table 2.

## Association between BCSCs expression with clinicopathological parameters

We identified several BCSCs in this review, which included SOX2 (Sry related HMG-Box 2), SOX4 (Sry related HMG-Box 4), ALDH1(Aldehyde-dehydrogenase 1), CD44, Nanog, Cripto-1 (Cysteine rich dommain), OCT4 (Octamer Binding Transcription Factor 4), CD133 (Prominin 1), β-arrestin-1 (ARRB1) and β-arrestin-2 (ARRB2), IGF1R, p-TFCP2L1 (Transcription Factor CP2-like Protein 1), and CDK1 (Cyclin Dependent Kinase 1). Eight out of 12 studies included performed Kaplan-Meier survival analysis to compare the effect of respective BCSCs expression on clinicopathological parameters. All of the genes were found to be significant prognostic factors based on univariate analysis. Moreover, multivariate analysis using Cox regression also showed that the majority of gene expression were independent prognostic factors; thus, it may play a role as a potentially valuable marker in predicting the recurrence-free, metastasis-free, or disease-free (recurrence/metastasis-free) with P<0.05 Table 3.

The expression of SOX 2 was significantly correlated with poorer recurrence free prognosis in the studies by Chiu (P = 0.0062 Univariate and P = 0.0029 Multivariate) and Ruan et al. (P = 0.001 Univariate and P = 0.029 Multivariate) [6, 17], Similarly, ALDH1 was also shown to be significantly associated with poorer recurrence free survival with a univariate P value of 0.04 and a multivariate P value of 0.001 from the studies by Xu and Senol et al.

**Table 2. Study characteristics.**

| Study (year) | Design | Participants | Control | Intervention | Gene Expression Analysis | Outcomes | | Mean/ median follow-up |
|---|---|---|---|---|---|---|---|---|
| | | | | | | Recurrence | Metastasis | |
| Ruan et al., 2012 [6] | Case-control | 32 BCa tissues | 32 corresponding normal tissues | - | qRT-PCR IHC | High SOX2 expression significantly played a role in predicting the recurrence-free survival in T1 BCa patients | - | 2 years |
| Keymoosi et al., 2013 [13] | Prospective Cohort | 159 patients with urothelial carcinoma | - | TURBT with no prior chemotherapy or radiation therapy | IHC on TMA slides | High ALDH1 and CD44 expressions were correlated with a significantly increased rate of recurrence (P = 0.013) | - | 46 months |
| Wei et al., 2015 [14] | Prospective cohort | 130 BCa patients | - | Cystectomy or Transurethral resection of the bladder tumor. | qRT-PCR IHC | High Cripto-1 was significantly associated with expression and tumor recurrence or metastasis (P = 0.007) | - | Not available |
| Senol et al., 2015 [15] | Prospective cohort | 163 cases of urothelial carcinomas of the bladder (UCB) | - | - | IHC | ALDH1 expression was significantly associated with disease recurrence (P<0.001), however, CD44 was not significantly associated (P = 0.688) | - | 23.60 ±16.88 months |
| Sedaghat et al., 2016 [2] | Retrospective cohort | 140 tissues from transitional cell carcinoma samples | - | - | TMA-based IHC | OCT4 expression had no correlation with tumor recurrence (P = 0.32) or CD133 (p = 0.71) | - | Not available |
| Siddiqui et al., 2019 [16] | Prospective cohort | 112 histopathologically proven BCa | | Bacillus Calmette Gurein (BCG), non-BCG, radical cystectomy with and without adjuvant therapy | IHC | High CD44 and NANOG expression were significantly associated with lower tumor recurrence (P<0.001) | - | Not available |
| Chiu et al., 2020 [17] | Retrospective cohort | For patients with transitional cell carcinoma of the urinary bladder, sample size not specified | - | - | IHC staining with SOX2 antibody | High SOX2 and IGF1R expression was correlated with poor recurrence-free survival and was increased in "poorly differentiated" malignant grade tumors (P = 0.0187) | - | Not available |
| Shen et al., 2015 [1] | Retrospective cohort | 309 patients with transitional cell carcinoma of the urinary bladder | - | - | IHC | High Sox4 expression was significantly associated with higher tumor grade (more likely to recurrent). (P = 3.71E-10) | High Sox4 expression was significantly associated with invasiveness (more likely to spread to other parts of the body). (P = 7.00E-04) | Not available |

(*Continued*)

**Table 2.** (Continued)

| Study (year) | Design | Participants | Control | Intervention | Gene Expression Analysis | Outcomes | | Mean/median follow-up |
| --- | --- | --- | --- | --- | --- | --- | --- | --- |
| | | | | | | Recurrence | Metastasis | |
| Xu et al., 2015 [18] | Prospective Cohort | 227 patients with bladder urothelial cell carcinoma (118 non-invasive and 109 invasive) | - | 118 patients with non-invasive bladder carcinoma: 11 underwent radical cystectomy and 107 underwent intravesical chemotherapy after transurethral resection 109 patients with invasive disease: 69 underwent radical cystectomy. 20 underwent partial cystectomy; and 20 underwent transurethral resection | IHC using ALDH1A1 antibody and secondary antibody from EnVision System | ALDH1 expression was significantly associated with tumor recurrence ($P \leq 0.05$). | ALDH1 expression was significantly associated with lymph node ($P = 0.008$) and tumor distant metastases ($P = 0.018$) | 52-months |
| Heo J et al., 2020 [19] | Retrospective cohort | 400 patients with urothelial carcinoma | - | TURBT | IHC | p-TFCP2L1 and CDK1 expression were not associated with recurrence ($P = 0.563$) | High levels of co-expression of p-TFCP2L1 and CDK1 were associated with distant metastasis ($P = 0.442$) | Not available |
| Shaifei et al., 2019 [20] | Case-control | 472 bladder tumors | 16 matched adjacent non-cancerous normal tissue | TURBT with no prior neoadjuvant treatment before surgery | IHC on TMA slides | DCLK1 expression was not associated with recurrence ($P = 0.314$) | DCLK1 expression was significantly associated with distant metastasis ($P = 0.042$) | Not available |
| Kallifatidis et al., 2019 [21] | Retrospective cohort | 43 bladder tumors in cohort 1; 43 bladder tumors in cohort 2 | 20 normal bladder | Cohort 2 receiving gemcitabine + cisplatin | qRT-PCR | | ARRB1 transcript levels in bladder tumor specimens from patients who developed metastasis were 7.7-fold elevated compared to the normal bladder and 5.2-fold elevated compared to BCa specimens from patients who did not develop metastasis | Not available |

ICH, Immunohistochemistry; NMIBC, non-muscle-invasive bladder cancer;(tissue microarray); qRT-PCR, Real-Time Quantitative Reverse Transcription Polymerase Chain Reaction; TURBT, Transurethral Resection of Bladder Tumor

respectively [15, 18]. In contrast, studies assessing the expression of CD44 with recurrence free survival reported conflicting findings, with Siddiqui et al. reporting univariate and multivariate P values of <0.001 whereas Senol et al. reported a multivariate P value of 0.074 [16].

**Table 3. Univariate and multivariate analysis of recurrence and metastasis as prognostic factors in patients with bladder carcinoma.**

| Outcome | Gene expression | Study (year) | Univariate analysis | | Multivariate analysis | |
|---|---|---|---|---|---|---|
| | | | HR (95% CI) | P value | HR (95% CI) | P value |
| **Recurrence-free** | SOX2 | Chiu (2020) | 2.467 (1.292–4.709) | 0.0062 | 2.966 (1.451–6.064) | 0.0029 |
| | | Ruan (2013) | 4.2 (1.827–9.654) | 0.001 | 3.187 (1.130–8.990) | 0.029 |
| | ALDH1 | Xu (2015) | 2.84 (1.19–7.14) | 0.040 | - | - |
| | CD44/Nanog | Siddiqui (2019) | 32.52 (9.79–107.99) | <0.001 | 25.45 (6.71–96.50) | <0.001 |
| | ALDH1 | Senol (2015) | - | - | 4.590 (2.042–10.319) | 0.001 |
| | CD44 | | - | - | 0.548(0.283–1.059) | 0.074 |
| **Metastasis** | ARRB1 | Kallifatidis (2020) | 1.35 (1.06–1.71) | 0.0137 | 1.07 (1.01–1.13) | 0.015 |
| | ARRB2 | | 0.03 (0.35–0.003) | 0.005 | 0.13 (0.86–0.02) | 0.006 |
| **Disease-free (recurrence/metastasis)** | Cripto-1 | Wei (2015) | 2.678 (1.280–5.605) | 0.009 | 2.306 (1.055–5.039) | 0.036 |
| | DCLK1 | Shafiei (2019) | 1.642 (1.063–2.534) | 0.025 | 1.564 (1.004–2.434) | 0.048 |

With regards to incidence of metastasis, Kallifatidis et al. reported that expressions of both ARRB1 and ARRB2 were significantly associated with increased metastasis with univariate findings of P = 0.0137 and P = 0.005 and multivariate findings of P = 0.015 and P = 0.006 respectively [21].

Both recurrence and metastasis were significantly marked in patients expressing Cripto-1 in a study by Wei et al. with results from univariate analysis showing P = 0.009 and multivariate analysis P = 0.036 [14]. Along the same line, DCLK1 was also demonstrated to be significantly associated with increased recurrence and metastasis with univariate and multivariate results showing P = 0.025 and P = 0.048 respectively in a study by Shaifei et al. [20].

## Discussion

Cancer cells utilize normal stem cell self-renewal for long-term proliferation and tissue-repair pathways for invasion; therefore, CSCs expression in cancer may be associated with disease prognosis and treatment outcomes [1]. Association between tumor biology and CSCs has been addressed in various types of cancer, including breast cancer, colorectal cancer, and bladder cancer. Many studies have revealed that CSCs were considered as an important factor leading to tumor recurrence and metastatic; however, its exact mechanisms are still unclear and may have a different pathway one to another [22]. By identifying and understanding the molecular mechanism of recurrent and metastatic BCa, numerous stem cell phenotypes, such as beta arrestins, SOX2, SOX4, transcription factor CP2 like 1 (TFCP2L1), and doublecortin-like kinase 1 (DCLK1), have all been identified and described.

Stem Cells govern homeostasis in human tissue. Specific gene expressions are regulated by transcription factors (TFs) and chromatin regulatory proteins. Embryonic stem cells express TFs such as Octamer-binding transcription factor 4 (OCT-4), Nanog Homeobox (Nanog), SRY-box2 (SOX-2), TFCP2L1, and Sry-Related HMG-BOX-4 (SOX4). These TFs are not expressed in already differentiated somatic cells; they suppress stem cells from differentiating. In patients with carcinoma, irregular gene activation causes stem cell proliferation. Cripto-1 is an example of an embryogenic gene overexpressed in bladder cancer.

Since multiple genes related to cancer are already identified in The Cancer Genome Atlas (TCGA), a model proposed to comprehend molecular characterization of muscle-invasive bladder cancer. Robertson et al. proposed a division between 5 subtypes of bladder cancer based on mRNA expression clustering: (1) luminal-papillary, (2) luminal-infiltrated, (3) luminal, (4) basal/squamous, (5) neuronal. Luminal papillary subtypes identified with FGFR3 mutations, TACC3 fusions, or papillary histology amplification. This subtype

responds poorly to cisplatin-based therapy. However, Tyrone kinase inhibitor of FGFR3 may be beneficial, as proved by early clinical trials. Luminal-infiltrated subtype identified with the expression of miR-200, EMT, and myofibroblast markers. This subtype may also be resistant to cisplatin-based therapy and thus can be beneficial as a negative predictive biomarker for chemotherapy response. The luminal subtype has a high expression of KRT20 and SNX31. The therapy targeting these specific mutation profiles may be beneficial. The basal-squamous subtype was identified with increased expression of CD274 (PD-L1) and CTLA4 immune markers. Therefore, immune checkpoint therapy and cisplatin-based therapy are appropriate options. The neuronal subtype is identified by the expression of neuronal genes and neuroendocrine. Etoposide-cisplatin therapy is proposed to be beneficial for this variant [23].

By identifying and understanding the molecular mechanism of recurrent and metastatic BCa, numerous stem cells phenotypes, such as beta arrestins, SOX2, SOX4, transcription factor CP2 like 1 (TFCP2L1), and doublecortin-like kinase 1 (DCLK1), have all been identified and described.

Kinase is an enzyme that catalyzes the transfer of phosphate groups. Mutation of kinase can cause cellular irregularities and lead to abnormal growth. Cyclin-dependent kinase 1 (CDK1) is a protein that functions as a threonine protein kinase for cell cycle regulation. However, CDK1 facilitates phosphorylation of TFCP2L1, activating embryonic stem cells in bladder cancer and driving tumorigenesis [19].

Two studies reporting a significant relationship between high expression of SOX2 with poor recurrence-free survival also found that SOX2 was highly expressed in tumors with poor pathological differentiation; thus, marking its role in BCa malignancy. SOX2 plays a role in promoting cell proliferation and enhancing cell survival during low-serum stress. BCa cancer cells' survival and spheroid-forming capability enhancement were induced by AKT phosphorylation due to IGF2/IGF1R induction, which was thought to be involved in molecular mechanism of SOX2 expression leading to poor tumor prognosis. Its mechanism made SOX2 a potential therapeutic target for BCa treatment [6, 17].

CD44 was one of the most stem cells which has been widely studied and was commonly expressed in BCa with a poor prognosis. Hu *et al*. [19] conducted a meta-analysis about the prognostic value of CD44 expression in BCa and found that CD44 expression may be associated with advanced T stage, tumor grade, and lymph node metastasis, but not with recurrence-free survival and overall survival of the disease. ALDH1 was also commonly reported to be a significant prognostic factor in tumor recurrence and metastasis. Xu *et al*. [18] and Senol *et al*. [15] further conducted univariate and multivariate survival analyses and also observed a statistically significant association between ALDH1 expression and recurrence-free survival (P<0.05). Moreover, Xu *et al*. also found that ALDH1 expression was related to distant tumor metastasis [18].

An inverse expression of ARRB1 and ARRB2 both significantly correlated with tumor metastasis. Kallifatidis *et al*. [21] conducted univariate and multivariate analysis and found that up-regulation of ARRB1 and down-regulation of ARRB2 both played a role as functional biomarkers to predict metastasis (P<0.05). Kallifatidis *et al*. [21] reported that ARRB2 negatively regulated the activation of STAT3, a transcription factor regulating the self-renewal nature of BCSCs. Conversely, ARRB1 was found to positively regulate BMI-1 and ARRB-1 were linked with poorer prognosis in BCa [21]. Overexpression of DCLK1 which was previously reported to be remarkable in cell progression and metastasis of colorectal cancer, was also found to be a significant prognostic factor on BCa. Multivariate Cox regression analyses conducted by Shafiei *et al*. [20] showed DCLK1 protein expression was an independent prognostic factor to poor disease-specific survival in BCa patients. However, the molecular

mechanism of the protein expression was not yet well-established. Cripto-1 or teratocarcinoma-derived growth factor-1 (TDGF-1) was found to have a significant association with tumor recurrence/metastasis in BCa patients (P = 0.007) and also as an independent prognostic factor identified with multivariate Cox regression analysis (P = 0.036), which validated its role to be a valuable marker as a disease-free predictor in BCa patients [14].

This systematic review showed that most BCSCs expressions were significantly associated with tumor recurrence and metastasis, suggesting its important role in patients' prognosis. CSC-specific cell-surface markers represent potential therapeutic targets. By knowing stem cell expression in BCa, therapeutic strategies could be set and implemented to improve disease outcomes. However, the mechanism of each CSCs was reported to be different due to its heterogeneity in the level of stem cells. Many cell-surface markers and signaling pathways are distinct in quiescent cells and proliferating cells; thus, microenvironmental interactions can alter stem cells' marker expression and signaling pathways [1]. Therefore, a major consideration for this approach remains the specificity of these markers [2]. In addition, specific CSC phenotypes appear to be correlated with disease outcomes, including risks of recurrence and metastasis [2]. These findings are comparable to those derived from another independent cohort of samples from the PanCancer Atlas which also shows that the expression of isolated markers is correlated with poor outcomes in bladder cancer [23]. Based on these findings, further studies regarding CSCs, especially their molecular mechanism, are warranted and may have significant contribution to the overall management of BCa.

Our study has several limitations. The majority of studies included did not show the mean or median follow-up time to determine the outcome. Each study also had different patients' characteristics, tumors' profiles, and treatment plans, which may also affect the recurrence and metastasis. We only presented a systematic review without further analysis; thus, we only can show that many studies have shown the beneficial impact of identifying BCSCs, and further studies are required.

Larger multicenter studies are needed to assess each factor that contributed to the recurrence and metastasis of BCa. However, statistical analysis sometimes has poor accuracy and is not applicable individually; thus, artificial intelligence has been further developed and may answer this problem. There have been several research that stated that artificial intelligence was believed to accurately predict cancer behavior, overall survival, and disease recurrence on BCa. Furthermore, Artificial intelligence can provide patient-tailored instruments for diagnosing and managing BCa [24].

## Conclusions

The detection of cancer stem cell expression offers a promising modality in predicting the prognosis of BCa. However, much is lacking in the molecular mechanisms underlying these processes. Hence, future research in this area is warranted and may highly contribute to the overall management of BCa.

## Supporting information

**S1 Checklist. PRISMA 2020 for abstracts checklist.**
(DOCX)

**S2 Checklist. PRISMA 2020 checklist.**
(DOCX)

## Author Contributions

**Conceptualization:** Agus Rizal Ardy Hariandy Hamid, Chaidir Arif Mochtar.

**Data curation:** Agus Rizal Ardy Hariandy Hamid.

**Formal analysis:** Agus Rizal Ardy Hariandy Hamid, Yasmina Zahra Syadza.

**Investigation:** Chaidir Arif Mochtar.

**Methodology:** Yasmina Zahra Syadza, Oliver Emmanuel Yausep, Roberto Bagaskara Indy Christanto.

**Project administration:** Bayu Hernawan Rahmat Muharia.

**Software:** Oliver Emmanuel Yausep, Roberto Bagaskara Indy Christanto.

**Supervision:** Agus Rizal Ardy Hariandy Hamid, Chaidir Arif Mochtar.

**Validation:** Agus Rizal Ardy Hariandy Hamid, Yasmina Zahra Syadza, Chaidir Arif Mochtar.

**Visualization:** Yasmina Zahra Syadza, Bayu Hernawan Rahmat Muharia.

**Writing – original draft:** Agus Rizal Ardy Hariandy Hamid, Yasmina Zahra Syadza, Oliver Emmanuel Yausep, Roberto Bagaskara Indy Christanto.

**Writing – review & editing:** Agus Rizal Ardy Hariandy Hamid, Chaidir Arif Mochtar.

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
