## [Decision Letter · Decision Letter 0]

14 Jun 2022

PONE-D-22-14251The expression of cancer stem cells and its effects on the propensity for recurrence and metastasis in bladder cancer: a systematic reviewPLOS ONE

Dear Dr. Hamid,

Thank you for submitting your manuscript to PLOS ONE. After careful consideration, we feel that it has merit but does not fully meet PLOS ONE’s publication criteria as it currently stands. Therefore, we invite you to submit a revised version of the manuscript that addresses the points raised during the review process.

Overall, this is a well-designed and well-written paper, but it needs serious revision, such as the following comments, where the main point in the discussion is the stem cell significance of bladder cancer and secondary points such as uniform expression.

We look forward to receiving your revised manuscript.

Kind regards,

Yun-Wen Zheng

Academic Editor

PLOS ONE

Journal Requirements:

Reviewers' comments:

Reviewer's Responses to Questions

**Comments to the Author**

1. Is the manuscript technically sound, and do the data support the conclusions?

Reviewer #1: Yes

Reviewer #2: Partly

2. Has the statistical analysis been performed appropriately and rigorously? 

Reviewer #1: N/A

Reviewer #2: Yes

3. Have the authors made all data underlying the findings in their manuscript fully available?

Reviewer #1: No

Reviewer #2: Yes

4. Is the manuscript presented in an intelligible fashion and written in standard English?

Reviewer #1: Yes

Reviewer #2: Yes

5. Review Comments to the Author

Reviewer #1: PONE-D-22-14251

General Comments

This manuscript is a systematic review of the prognostic significance of stem cell gene or cancer stem cell gene expression in bladder cancer. It appears to be just a summary of papers related to the topic, with no major flaws.

Major Concerns

Both tables span multiple pages and are difficult to understand at a glance. The tables need to be in the range of one or two pages.

Some of the cancer stem cell genes and stem cell genes in the paper are not very familiar. To be sure, all of the genes need to be explained and verified.

There is no uniformity in the words used within the tables. For example, is there any difference between qrtRT-PCR, qRT-PCR, and RT-q-PCR? It would be more helpful to the reader if the designations were unified whenever possible.

Some studies do not seem to correlate these stem cell gene expressions with prognostic factors, but the abstract reads as if there were significant differences in all studies.

The stem cell significance of bladder cancer should be discussed, as it has been mentioned in the recent TCGA studies (PMID: 24476821, 28988769, 29617660).

Reviewer #2: This manuscript describes the prognostic value of stem cell markers in Bladder Cancer. The authors find that the expression of stem cell markers was significantly associated with tumor recurrence and metastasis by systematic review, suggesting its important role in Bladder Cancer. Overall, this manuscript is well written and the logical is not bad. Most of the results can support their conclusions. However, the authors should consider validating their finding with an independent cohort of samples if possible. The manuscript is acceptable, but some minor revisions should be considered:

1. If the author could validate their finding with another independent cohort of samples to confirm, some of these results will be perfect. For example, according to the association between gene/protein expression and recurrence, analysis of TCGA (PanCancer Atlas) data is able to confirm that the expression of their isolated markers showed poor outcomes in bladder cancer. The authors can freely download and analyze the dataset from cBioportal (http://cbioportal.org).

2. “The expression of cancer stem cells” in the title is strange and overestimate. The authors only link the expression of stem cell markers, but not the expression of cancer stem cells, to poor clinical outcome. Please change the title.

6. PLOS authors have the option to publish the peer review history of their article (what does this mean?). If published, this will include your full peer review and any attached files.

Reviewer #1: No

Reviewer #2: No

---

## [Author Response · Author response to Decision Letter 0]

30 Nov 2022

Reviewer #1

Both tables span multiple pages and are difficult to understand at a glance. The tables need to be in the range of one or two pages.

• Thank you for your feedback, we have adjusted the table sizes accordingly whilst maintaining the amount of content within them

Some of the cancer stem cell genes and stem cell genes in the paper are not very familiar. To be sure, all of the genes need to be explained and verified.

• We have included a brief description of all stem cell genes

There is no uniformity in the words used within the tables. For example, is there any difference between qrtRT-PCR, qRT-PCR, and RT-q-PCR? It would be more helpful to the reader if the designations were unified whenever possible.

• The referred designations have been unified. Apologies for the earlier confusion as we tried to stick with the terms the authors used in their respective studies

Some studies do not seem to correlate these stem cell gene expressions with prognostic factors, but the abstract reads as if there were significant differences in all studies.

• Thank you for bringing this to our attention, we have reworded that section on the abstract accordingly

The stem cell significance of bladder cancer should be discussed, as it has been mentioned in the recent TCGA studies (PMID: 24476821, 28988769, 29617660).

• We appreciate this suggestion and have included the significance of stem cell in bladder cancer according to recent TCGA studies

Reviewer #2

1. If the author could validate their finding with another independent cohort of samples to confirm, some of these results will be perfect. For example, according to the association between gene/protein expression and recurrence, analysis of TCGA (PanCancer Atlas) data is able to confirm that the expression of their isolated markers showed poor outcomes in bladder cancer. The authors can freely download and analyze the dataset from cBioportal (http://cbioportal.org).

• Thank you for the additional sources of reference, we have incorporated these additional findings to support our discussion

2. “The expression of cancer stem cells” in the title is strange and overestimate. The authors only link the expression of stem cell markers, but not the expression of cancer stem cells, to poor clinical outcome. Please change the title.

• We thank you for bringing this to our attention. The appropriate title changes have been made

---

## [Decision Letter · Decision Letter 1]

3 Jan 2023

PONE-D-22-14251R1The expression of cancer stem cells and its effects on the propensity for recurrence and metastasis in bladder cancer: a systematic reviewPLOS ONE

Dear Dr. Hamid,

Thank you for submitting your manuscript to PLOS ONE. After careful consideration, we feel that it has merit but does not fully meet PLOS ONE’s publication criteria as it currently stands. Therefore, we invite you to submit a revised version of the manuscript that addresses the points raised during the review process.

I appreciate your efforts to improve the manuscript. After a second round of review, further comments are as follows, especially that similarities and differences regarding marker genes should be enhanced.   

We look forward to receiving your revised manuscript.

Kind regards,

Yun-Wen Zheng

Academic Editor

PLOS ONE

Reviewers' comments:

Reviewer's Responses to Questions

**Comments to the Author**

1. If the authors have adequately addressed your comments raised in a previous round of review and you feel that this manuscript is now acceptable for publication, you may indicate that here to bypass the “Comments to the Author” section, enter your conflict of interest statement in the “Confidential to Editor” section, and submit your "Accept" recommendation.

Reviewer #3: (No Response)

Reviewer #4: (No Response)

2. Is the manuscript technically sound, and do the data support the conclusions?

Reviewer #3: Yes

Reviewer #4: (No Response)

3. Has the statistical analysis been performed appropriately and rigorously? 

Reviewer #3: N/A

Reviewer #4: (No Response)

4. Have the authors made all data underlying the findings in their manuscript fully available?

Reviewer #3: Yes

Reviewer #4: (No Response)

5. Is the manuscript presented in an intelligible fashion and written in standard English?

Reviewer #3: Yes

Reviewer #4: (No Response)

6. Review Comments to the Author

Reviewer #3: In this manuscript, authors reviewed the prognoses of stem cell genes associated with bladder cancer (BCa). However, this review has the following shortcomings:

1. Are there any other potential stem cell genes related to the prognosis of BCa?

2. Do the prognoses of these stem cell genes have any difference in all and subtypes of BCa patients?

3. Please check the format of reference in the whole paper. For example, full stop in line 49 “…metastasize. [1-3] However, both…” is before or after [1-3]?

4. When author illustrated the data from table 3, they mentioned 13 stem cell genes in line 152-154. But why only 9 stem cell genes are shown in table 3?

5. It’s better to explain the full name of stem cell genes for the first time.

6. It’s better to state the results in detail.

7. In line 178, “…of In …” may be a typo mistake.

Reviewer #4: In this study, Hamid et al. performed a systematic review to synthesize the potential BCSCs prognostic factors for BCa and summarized that the different BCSCs makers might help predict the risks of metastasis and recurrence in BCa.

Specific comments:

1. The review may describe comprehensive data in the results section.

2. There was no evaluation and analysis to compare different markers of BCSCs in the same outcome group.

3. The relationship of these BCSCs makers should be discussed.

7. PLOS authors have the option to publish the peer review history of their article (what does this mean?). If published, this will include your full peer review and any attached files.

Reviewer #3: No

Reviewer #4: No

---

## [Author Response · Author response to Decision Letter 1]

11 Apr 2023

Response to Reviewers

Reviewer #3

1. Are there any other potential stem cell genes related to the prognosis of BCa?

• Dear reviewer, as of the writing of this paper all the identified stem cell genes that may be related to the prognosis of BCa have been included in this writing 

2. Do the prognoses of these stem cell genes have any difference in all and subtypes of BCa patients?

• Dear reviewer, to our knowledge no study has yet been performed on the effects of individual stem cell genes on each subtype of BCa patients 

3. Please check the format of reference in the whole paper. For example, full stop in line 49 “…metastasize. [1-3] However, both…” is before or after [1-3]?

• We place the reference numbers after full stops

4. When author illustrated the data from table 3, they mentioned 13 stem cell genes in line 152-154. But why only 9 stem cell genes are shown in table 3?

• Dear author we apologize for the confusion, this is because out of the 13 stem cell genes studied by the identified articles, only 9 of them performed univariate or multivariate analyses to compare in Table 3

5. It’s better to explain the full name of stem cell genes for the first time.

• Thank you for your feedback, we have added the full names of each stem cells accordingly 

6. It’s better to state the results in detail.

• Thank you for the feedback we have corrected accordingly

7. In line 178, “…of In …” may be a typo mistake.

• Thank you for the feedback we have corrected accordingly

Reviewer #4

1. The review may describe comprehensive data in the results section.

• We thank you for the feedback and have described the data in greater detail

2. There was no evaluation and analysis to compare different markers of BCSCs in the same outcome group.

• Yes we did not perform further analyses such as forest plots for this article

3. The relationship of these BCSCs makers should be discussed.

• We thank you for the feedback and have added the points accordingly

---

## [Decision Letter · Decision Letter 2]

28 Apr 2023

The expression of stem cells markers and its effects on the propensity for recurrence and metastasis in bladder cancer: a systematic review

PONE-D-22-14251R2

Dear Dr. Hamid,

We’re pleased to inform you that your manuscript has been judged scientifically suitable for publication and will be formally accepted for publication once it meets all outstanding technical requirements.

Kind regards,

Yun-Wen Zheng

Academic Editor

PLOS ONE

Additional Editor Comments (optional):

Reviewers' comments:

Reviewer's Responses to Questions

**Comments to the Author**

1. If the authors have adequately addressed your comments raised in a previous round of review and you feel that this manuscript is now acceptable for publication, you may indicate that here to bypass the “Comments to the Author” section, enter your conflict of interest statement in the “Confidential to Editor” section, and submit your "Accept" recommendation.

Reviewer #3: All comments have been addressed

Reviewer #4: (No Response)

2. Is the manuscript technically sound, and do the data support the conclusions?

Reviewer #3: Yes

Reviewer #4: (No Response)

3. Has the statistical analysis been performed appropriately and rigorously? 

Reviewer #3: Yes

Reviewer #4: (No Response)

4. Have the authors made all data underlying the findings in their manuscript fully available?

Reviewer #3: Yes

Reviewer #4: (No Response)

5. Is the manuscript presented in an intelligible fashion and written in standard English?

Reviewer #3: Yes

Reviewer #4: Yes

6. Review Comments to the Author

Reviewer #3: (No Response)

Reviewer #4: (No Response)

7. PLOS authors have the option to publish the peer review history of their article (what does this mean?). If published, this will include your full peer review and any attached files.

Reviewer #3: No

Reviewer #4: No

---

## [Editor Report · Acceptance letter]

7 May 2023

PONE-D-22-14251R2 

The expression of stem cells markers and its effects on the propensity for recurrence and metastasis in bladder cancer: a systematic review 

Dear Dr. Hamid:

I'm pleased to inform you that your manuscript has been deemed suitable for publication in PLOS ONE. Congratulations! Your manuscript is now with our production department. 

Kind regards, 

on behalf of

Dr. Yun-Wen Zheng 

Academic Editor

PLOS ONE